# High-Sensitivity Goos-Hänchen Shifts Sensor Based on BlueP-TMDCs-Graphene Heterostructure

**DOI:** 10.3390/s20123605

**Published:** 2020-06-26

**Authors:** Lei Han, Zhimin Hu, Jianxing Pan, Tianye Huang, Dapeng Luo

**Affiliations:** School of Mechanical Engineering and Electronic Information, China University of Geosciences (Wuhan), Wuhan 430074, China; hanlei@cug.edu.cn (L.H.); huzhimin@cug.edu.cn (Z.H.); jianxing_pan@163.com (J.P.); tianye_huang@163.com (T.H.)

**Keywords:** surface plasmon resonance, Goos–Hänchen shifts, blue phosphorene, transition metal dichalcogenides, graphene, sensitivity

## Abstract

Surface plasmon resonance (SPR) with two-dimensional (2D) materials is proposed to enhance the sensitivity of sensors. A novel Goos–Hänchen (GH) shift sensing scheme based on blue phosphorene (BlueP)/transition metal dichalogenides (TMDCs) and graphene structure is proposed. The significantly enhanced GH shift is obtained by optimizing the layers of BlueP/TMDCs and graphene. The maximum GH shift of the hybrid structure of Ag-Indium tin oxide (ITO)-BlueP/WS_2_–graphene is −2361λ with BlueP/WS_2_ four layers and a graphene monolayer. Furthermore, the GH shift can be positive or negative depending on the layer number of BlueP/TMDCs and graphene. For sensing performance, the highest sensitivity of 2.767 × 10^7^λ/RIU is realized, which is 5152.7 times higher than the traditional Ag-SPR structure, 2470.5 times of Ag-ITO, 2159.2 times of Ag-ITO-BlueP/WS_2_, and 688.9 times of Ag-ITO–graphene. Therefore, such configuration with GH shift can be used in various chemical, biomedical and optical sensing fields.

## 1. Introduction

The Goos-Hänchen (GH) shift refers to the lateral spatial shift of the center of mass of the bounded beam relative to the geometric prediction [1]. The GH migration results from the role dispersion of the Fresnel reflection coefficient [2,3]. When the phase of reflection coefficient changes significantly near the critical angle of total reflection, GH effect can be enhanced [4,5,6]. Due to the advantages of GH shift in precision measurement and optical sensing, new attention has been observed [7,8,9]. In the fields of optics, chemistry and sensors, GH shift has been widely discussed [10,11] and there is always a goal for researchers to obtain large GH shifts.

The GH shift at the interface of two homogeneous materials with different optical properties is usually very small, almost equal to the incident wavelength [12]. Leveraging on metal to excite surface plasmon polaritons (SPP) is an effective way to improve GH shift [13]. SPPs are kinds of vertically constrained evanescent electromagnetic waves [14,15]. According to Snell’s law, if the incident angle is larger than the total reflection angle, the total reflection phenomenon will appear when a beam of light is transmitted from a dense medium to a sparse medium [16]. When total reflection occurs, if there is no energy loss, it is called total internal reflection (TR), if there is energy loss, it is called attenuated total reflection (ATR). From the point of view of physical optics, a more in-depth study of total reflection shows that when total reflection occurs, the beam enters a wavelength-level depth in the optical medium, and its amplitude decays exponentially along the direction perpendicular to the interface. At the same time, in the incident plane, it transmits for a certain distance along the interface direction, and then returns to the optical dense medium. From the point of view of physical optics, a more in-depth study of total reflection shows that when total reflection occurs, the beam enters a wavelength level depth in the optical medium, and its amplitude decays exponentially along the direction perpendicular to the interface. At the same time, in the incident plane, it transmits for a certain distance along the interface direction, and then returns to the optical dense medium [17]. You et al. proposed a long-range surface plasmon resonance (SPR) mode, in which gold (Au) is the excitation layer of SPPs, so as to improve the GH shift [18]. The sliver (Ag) film layer of SPR can easily adjust the position of the minimum reflection and the maximum GH shift [19]. The sensitivity of GH shift to the refractive index of the surrounding medium is obtained by optimizing the incident angle and the thickness of Au [20].

Compared with other metal oxides, Indium tin oxide (ITO) has the characteristics of anti-corrosion, high transmittance, good conductivity, etc., which are widely used in the field of optical sensing [21,22]. The GH shift of p-polarized laser beam reflected from ITO surface under complex field has been studied [23]. In addition, two-dimensional (2D) nanomaterial with high thermal conductivity, carrier mobility, wide band optical response spectrum and strong nonlinear optical properties have attracted more and more attention [24,25]. The 2D materials include graphene [26,27], black phosphorus (BP) [28,29], and transition metal dichalcogenides (TMDCs) [30,31]. Graphene is the most representative 2D material widely used in optical sensors [32]. Luo studied the electro-optical modulation and magneto-optical modulation of GH shift in the double graphene coated waveguide [33]. Zhou et al. researched the GH effect in graphene substrate system by transfer matrix method [34]. Zhao et al. theoretically proposed the GH shift of light beam in a defect photonic crystal composed of dielectric multilayer and graphene [35]. By optimizing the thickness and layers number of the Au-MoS_2_–graphene hybrid, You et al. gained the highest GH shift of 235.8λ [36]. Through the optimization and comparative analysis of the Au-ITO-TMDCs–graphene hybrid structure, Han et al. obtained the highest GH shift of 801.7λ with MoSe_2_ monolayer and graphene bilayer [37]. Zhu et al. discovered another promising 2D material, Blue phosphorene (BlueP) [38]. The BlueP has the same thermal stability as BP with a band width of 2 eV, so it has a broad application prospect for sensing applications [39]. In addition, because the monolayer of BlueP and TMDCs have the same hexagonal crystal structure, it is easy to construct the heterstructure of BlueP/TMDCs [40]. Srivastava obtained the SPR sensor of BlueP/MoS_2_ heterostructure to improve the sensitivity [41]. Sharma et al. proposed to use the BlueP/TMDCs molybdenum disulfide heterostructure, and compared with the traditional graphene SPR sensors [42].

In this paper, the hybrid structure of BlueP/TMDCs (Blue/WS_2_, BlueP/MoS_2_, BlueP/MoSe_2_, BlueP/WSe_2_) and graphene coated with ITO and Ag thin film is proposed. The maximum GH shift achieved was −2361λ for four layers BlueP/WS_2_ and monolayer graphene. In addition, the highest sensitivity for index sensing reached 2.767 × 10^7^λ/RIU, which is 5152.7 times higher than the traditional Ag structure. We believe that this scheme with 2D materials has potential in highly sensitive sensors.

## 2. Design Consideration and Mathematical Model

The hybrid structure of Ag-ITO-BlueP/TMDCs–graphene based on the Kretschmann structure is shown in Figure 1. The *p*-polarized He-Ne laser emitted at 632.8 nm is collimated by a Glan–Taylor prism. Under the Kretschmann structure, the glass slide coated with metal film is fixed on the base of equilateral prism made of high refractive index (RI) glass with refractive index matching solution [43]. The incident light is irradiated on the SPR sensor through the side of the equilateral triangular coupling prism. The prism coupling device is controlled by a mobile rotary table, so as to change the angle of the incident light.

In the following description of the refractive index (RI) in each layer, λ is the wavelength of the incident light, and its unit is um. In the first layer, the SF11 prism with RI (*n*_1_) is obtained [43]:(1)$$n_{1}=\sqrt{\frac{1.73759695\lambda^{2}}{\lambda^{2}-0.013188707}+\frac{0.313747346\lambda^{2}}{\lambda^{2}-0.0623068142}+\frac{1.89878101\lambda^{2}}{\lambda^{2}-155.23629}+1}$$

Then, in the second layer, BK7 glass with RI (*n*_2_) is obtained [44]:(2)$$n_{2}=\sqrt{\frac{1.03961212\lambda^{2}}{\lambda^{2}-0.00600069867}+\frac{0.231792344\lambda^{2}}{\lambda^{2}-0.0200179144}+\frac{1.01046945\lambda^{2}}{\lambda^{2}-103.560653}+1}$$

The third layer is Ag thin film and its RI (*n*_3_) is obtained through the Drude model [45]:(3)$$n_{3}=\sqrt{1-\frac{\gamma_{c}\lambda^{2}}{\gamma_{p}^{2}(\gamma_{c}+i\lambda )}}=\sqrt{1-\frac{17.6140\lambda^{2}}{0.14541^{2}\times \left(17.6140+i\lambda \right)}}$$

The ITO film as fourth layer with RI (*n*_4_) is [46]:(4)$$n_{4}=\sqrt{3.8-\frac{\gamma_{c}\lambda^{2}}{\gamma_{p}^{2}(\gamma_{c}+i\lambda )}}=\sqrt{3.8-\frac{11.2107\lambda^{2}}{0.56497^{2}\times \left(11.2107+i\lambda \right)}}$$

Subsequently, the 2D material of BlueP/TMDCs and graphene with monolayer and RI is shown as Table 1 [47,48].

The sensing medium is used for deionized (DI) water and its RI (*n*_7_) is obtained [43]:(5)$$n_{7}=\sqrt{\sum_{i=1}^{4}\frac{A_{i}\lambda^{2}}{\lambda^{2}-t_{i}^{2}}+1}$$
where the Sellmeier coefficients *A*_1_ = 0.5666959820, *A*_2_ =0.1731900098, *A*_3_ =0.02095951857, *A*_4_ = 0.1125228406, *t*_1_= 0.005084151894, *t*_2_ = 0.01818488474, *t*_3_ = 0.02625439472, *t*_4_ = 10.73842352.

Therefore, the RI is *n*_1_ = 1.7786, *n*_2_ = 1.5151, *n*_3_ = 0.1350 + 3.9850i, *n*_4_ = 1.858 + 0.058i, *n*_7_ = 1.332 + *n_bio_*. The *n_bio_* represents the RI change of DI water. The thickness of BK7 glass and sensing medium is both 100 nm. In order to compare the properties of 2D materials, we set the thickness of Ag and ITO to 45 nm and 10 nm, respectively. When the number of graphene layers n ≤ 5, it is reasonable to treat monolayer as a non-interacting [49]. Hence, we only use graphene and BlueP/TMDCs with 5 layers or less, and ignore the interaction between them.

The thickness and dielectric constant of each layer are set as *d_k_* and *ε_k_* (*ε_k_* = *n_k_*^2^) (*k* = 1,2,…7). In order to analyze the reflectivity (*R_p_*) and phase (*ψ_p_*), the transfer matrix method (TMM) and the Fresnel equation based on *n*-layer model are used [47]. The SPR sensor is composed of parallel stacking in Z direction perpendicular to the sensing interface. The *M* is the structure of the transmission matrix (TM), and *p*-polarized light is gained through the following relationship [43]:(6)$$M=\prod_{k=2}^{N-1}M_{k}=\prod_{N}^{N-1}\left[\begin{array}{cc} \cos\alpha_{k} & (-i\sin\alpha_{k})/p_{k}\\ -ip_{k}\sin\alpha_{k} & \cos\alpha_{k} \end{array}\right]=[\begin{array}{l} M_{11}\\ M_{21} \end{array}\begin{array}{l} M_{12}\\ M_{22} \end{array}]$$
where
(7)$$p_{k}=\sqrt{\left(\frac{\nu_{k}}{\varepsilon_{k}}\right)}\cos\theta_{k}=\frac{\sqrt{\varepsilon_{k}-n_{1}^{2}\sin^{2}\theta_{1}}}{\varepsilon_{k}}$$
and
(8)$$\alpha_{k}=\frac{2\pi d_{k}}{\lambda }\sqrt{\varepsilon_{k}-n_{1}^{2}\sin^{2}\theta_{1}}$$

The total reflection coefficient (*r_p_*) of *p*-polarized light is related to the matrix as follows:(9)$$\gamma_{p}=\frac{\left(M_{11}+M_{12}p_{N}\right)p_{1}-(M_{21}+M_{22}p_{N})}{\left(M_{11}+M_{12}p_{N}\right)p_{1}+(M_{21}+M_{22}p_{N})}$$
where *p*_1_ corresponds to the SF11 prism layer and *p*_7_ to the water layer. The *R_p_* and *ψ**_p_* of the *p*-polarized light are shown as [17]:(10)$$R_{p}={\left|\gamma_{p}\right|}^{2}$$
(11)$$\psi_{p}=\arg(\gamma_{p})$$

We can use the fixed phase method to calculate the GH shift as followed:(12)$$S=-\frac{1}{k_{0}}\frac{d\psi_{p}}{d\theta_{1}}=-\frac{\lambda }{2\pi }\frac{d\psi_{p}}{d\theta_{1}}$$
where the *θ*_1_ is the incident angle.

## 3. Result and Discussion

As shown in Figure 2, the reflectance, phase and GH shift of conventional Ag and Ag-ITO structure are compared and analyzed. The reflectivity (red dot line) and phase (blue dot line) are shown in Figure 2a. The SPR curve shows that there is a narrow reflection angle near 52.71° and 55.22° respectively, and the minimum reflectivity is 0.068 a.u for Ag structure and 0.014 a.u for Ag-ITO structure, respectively. In Figure 2b, the highest GH shift with Ag = 45 nm is 60.21λ. With Ag = 45 nm and ITO = 10 nm, the maximum GH shift attains 62.11λ indicating ITO can increase the GH shift and other performance.

Although ITO plays a certain role in the increase of GH shift, the enhancement is still relatively small. Next, we study the influence of the graphene layer, as shown in Figure 3. For the graphene monolayer, the reflectivity is 0.0024 a.u at 55.44° and the GH shift is 125.7λ. When the graphene bilayer is added to the Ag-ITO structure, the best performance is obtained with the largest GH shift of −439.7λ. We can observe that the phase change from Z-shaped-like to Lorentzian-like, and GH shift change from positive to negative. Then, with the increase of graphene layers, the GH shift is −67.56λ, −33.46λ, −20.19λ, respectively.

Similarly, the BlueP/TMDCs is added to the Ag-ITO structure, as shown in Figure 4. In Table 2, the optimal GH shift (S/λ) with different number of BlueP/TMDCs is obtained. Hence, In the BlueP/TMDCs, we can understand that the BlueP/MoSe_2_ has the greatest contribution to Ag-ITO structure.

From Figure 2, Figure 3 and Figure 4, there are four important features. First, at a certain thickness of Ag-ITO film, due to the large real-part value of BlueP/TMDCs and graphene dielectric function, with the increase of BlueP/TMDCs and graphene layers, the SPR resonance angle has a large red shift. Second, the imaginary part of dielectric function of graphene layer is larger than that of BlueP/TMDCs layer, which leads to a large loss of electronic energy. Third, the resonance depth (i.e., the minimum reflectivity) strongly depends on the number of BlueP/TMDC and graphene layers deposited on Ag-ITO films. The light energy absorbed by Ag-ITO film is not enough to excite strong SPR. By further coating BlueP/TMDCs and graphene on the surface of Ag-ITO film, the light absorption of the hybrid structure can be enhanced effectively, thereby promoting a stronger SPR excitation.

Figure 5 shows the GH shift relative to the incident angle when graphene is monolayer and the number of BlueP/TMDCs layers changes from monolayer to five layers. The optimal GH shift and resonance angle with different number of BlueP/TMDCs and graphene monolayer are obtained as Table 3. We know that with the increase of the optimal GH shift of each BlueP/TMDCs, the corresponding resonance angle also increases.

Subsequently, the BlueP/TMDCs monolayer and different number of graphene layers are added to the Ag-ITO structure of SPR, as shown in Figure 6. The optimal GH shift with a different number of graphene and BlueP/TMDCs monolayer are obtained, as shown in Table 4.

In Table 5, the optimal GH shift with a different number of BlueP/TMDCs and graphene layers are summarized, where the bold indicates the highest GH shift value under the structure. In the BlueP/MoS_2_ and graphene, the highest GH shift is −385.8λ in BlueP/MoS_2_ bilayer and graphene monolayer. Then, the maximum GH shift with BlueP/WS_2_ four layers and graphene monolayer is -2361λ. Subsequently, the largest GH shift with BlueP/WSe_2_ three layers and graphene monolayer of -655.5λ is obtained. Finally, the highest GH shift of 456.9λ is obtained for both BlueP/MoSe_2_ and graphene monolayer. Therefore, in the Ag-ITO-BlueP/TMDCs–graphene structure, BlueP/WS_2_ has the greatest contribution to GH shift. The monolayer of graphene has the best performance.

In our study, the RI of the sensing medium (∆*n*_7_) is changed, then and the GH shift is shown a giant red shift. Hence, the proposed new SPR heterostructure is used as a high sensitivity sensor based on shift variation. ∆*GH* is defined as highest value of the varying GH shift and the sensitivity is defined as *S’_p_* = ∆*GH*/∆*n*_7_. We define the traditional SPR Ag, Ag-ITO, Ag-ITO-BlueP/WS_2_ (monolayer), Ag-ITO–graphene (monolayer), Ag-ITO-BlueP/WS_2_ (monolayer)–graphene (monolayer), Ag-ITO-BlueP/WS_2_ (four layers)–graphene (monolayer) structure as Structure I to Structure VI and show as Figure 7. In Table 6, the optimal *S’_P_* with Δ*GH* and ∆*n*_7_ for Structure I to Structure VI are gained. Therefore, Structure VI is 5152.7 times higher than Structure I, 2470.5 times higher than Structure II, 2159.2 times higher than Structure III, and 688.9 times higher than Structure IV.

Better compared with previous research results, Table 7 summarizes the GH shift and sensitivity based on SPR sensor. In references [9], the GH shift of 12.5λ is obtained by traditional Au thin film. In reference [50], when MoS_2_ of 2D material and air was added to the SPR biosensor, the GH shift was improved to 40.5λ. We can find that 2D material and air can improve the GH shift of SPR sensor. In reference [51], when the graphene replaced MoS_2_, the GH shift increased to 61.1λ. Therefore, compared with MoS_2_, graphene improves the performance of the SPR sensor more significantly. However, when graphene and MoS_2_ were added to the Au film of SPR sensor, the GH shift increased to 235.8λ for reference [36], and the highest sensitivity was obtained as 5.545 × 10^5^λ/RIU. In reference [37], when the ITO and MoSe_2_ replaced MoS_2_, the GH shift increased to 801.7λ, and the maximum sensitivity was 8.02 × 10^5^λ/RIU. In this work, we use BlueP/TMDCs instead of TMDCs, and change metal into Ag, so as to construct the SPR sensor with Ag-ITO- BlueP/WS_2_–graphene hybrid structure. The optimal GH displacement is 2361λ and the maximum sensitivity is 2.767 × 10^7^λ/RIU. Based on the analysis, we can see that our novel SPR sensor improves the GH shift and sensitivity significantly.

## 4. Conclusions

In this paper, the GH shift of the Kretschmann configuration combined with SPR-based 2D nanomaterials is studied. When SPPs were excited, we theoretically proved the influence of the number of graphene and BlueP/TMDCs layers on the GH shift, and obtained a huge GH shift by using the mixed structure of BlueP/WS_2_ four layers and graphene monolayer. The maximum GH shift is 2361 times that of the incident wavelength. Compared with the traditional SPR structure, the shift of the structure is increased by more than 39.21 times. In addition, by changing the number of BlueP/TMDCs layers, we can control the positive and negative shift of GH in the structure of BlueP/TMDCs–graphene. The maximum GH shift corresponding to the highest sensitivity is 2.767 × 10^7^λ/RIU, which is 5152.7 times higher than the traditional SPR of Ag, 2462.8 times of Ag-ITO, 2159.2 times of Ag-ITO-BlueP/WS_2_, and 688.9 times of Ag-ITO–graphene. The sensing layer we use is deionized water, therefore, it is suitable as a sensing medium with a refractive index close to 1.332, to gain a higher sensitivity. This structure is expected to be a candidate for high-performance sensors.

## Figures and Tables

**Figure 1 sensors-20-03605-f001:**
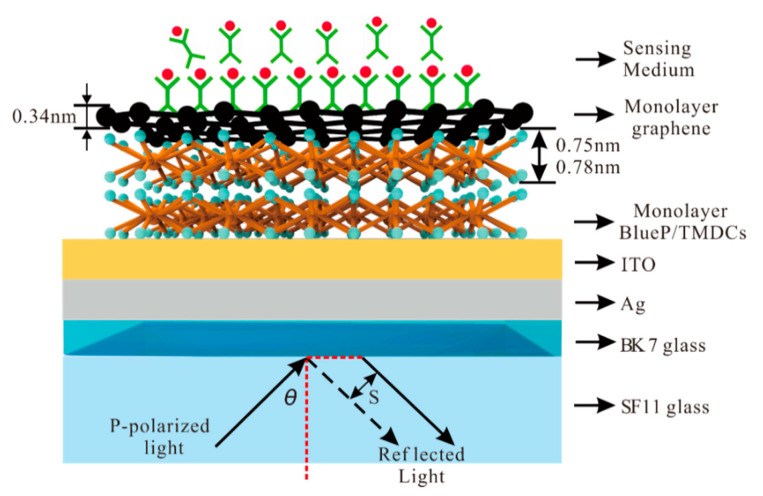
Schematic diagram of Ag- Indium tin oxide (ITO)-Blue Phosphorene (BlueP)/transition metal dichalogenides (TMDCs)–graphene enhanced Goos–Hänchen (GH) shift of surface plasmon resonance (SPR) sensor.

**Figure 2 sensors-20-03605-f002:**
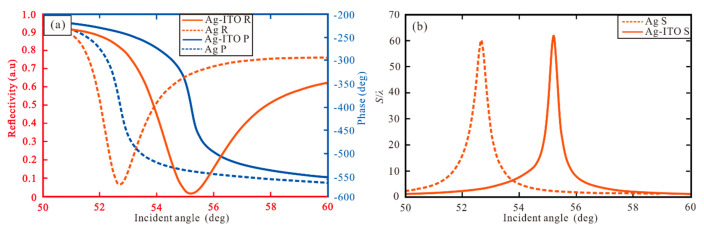
(**a**) Reflectivity and phase, and (**b**) GH shift for Ag and Ag-ITO structure with incident angle.

**Figure 3 sensors-20-03605-f003:**
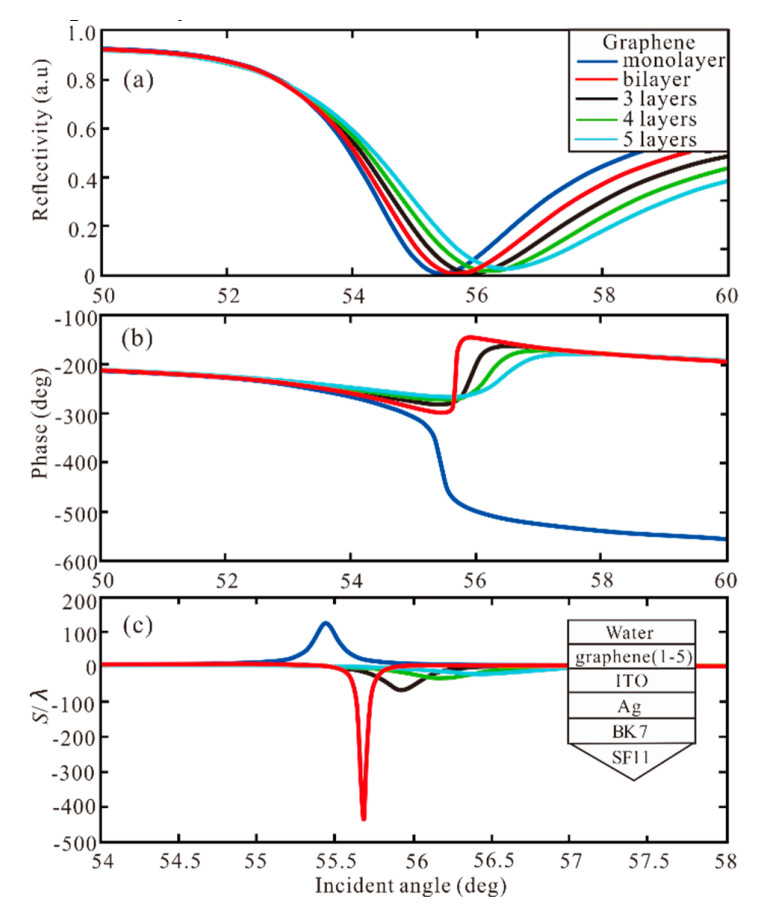
(**a**) Reflectivity, (**b**) Phase and (**c**) GH shift with incident angle for different number of graphene layers.

**Figure 4 sensors-20-03605-f004:**
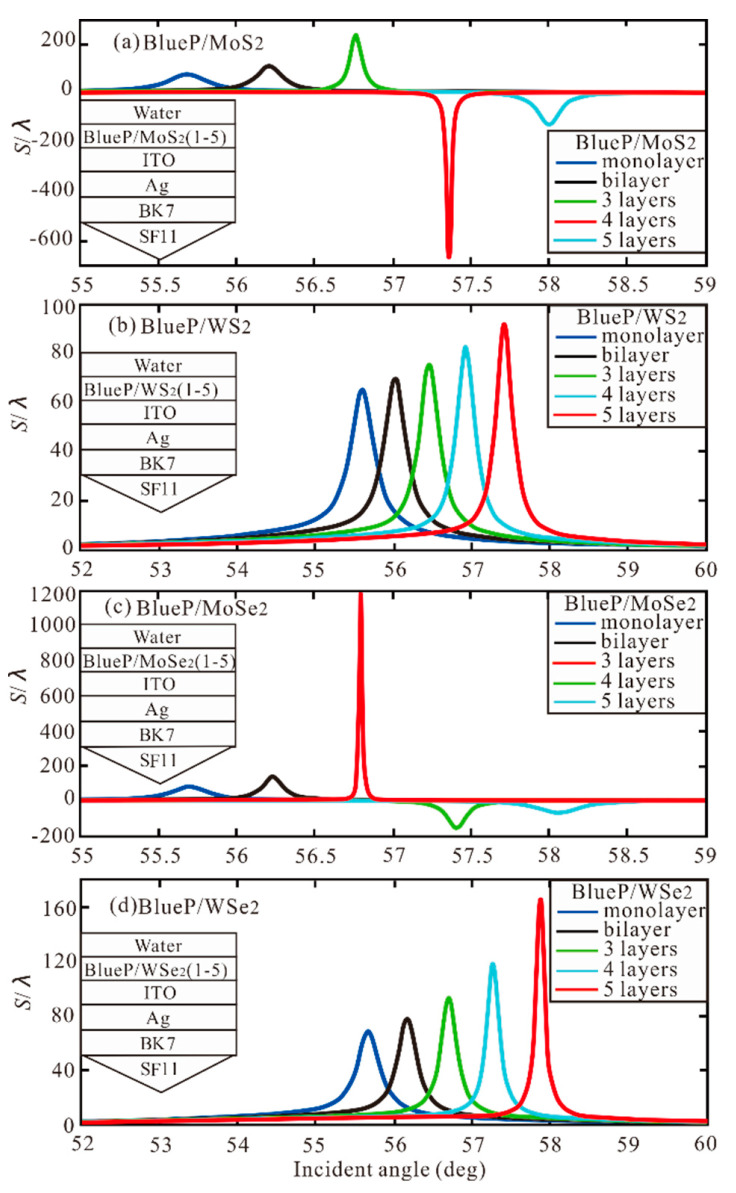
The GH shift with incident angle for different number of BlueP/TMDCs (**a**) BlueP/MoS_2_ layers, (**b**) BlueP/WS_2_ layers, (**c**) BlueP/MoSe_2_ layers, (**d**) BlueP/WSe_2_ layers.

**Figure 5 sensors-20-03605-f005:**
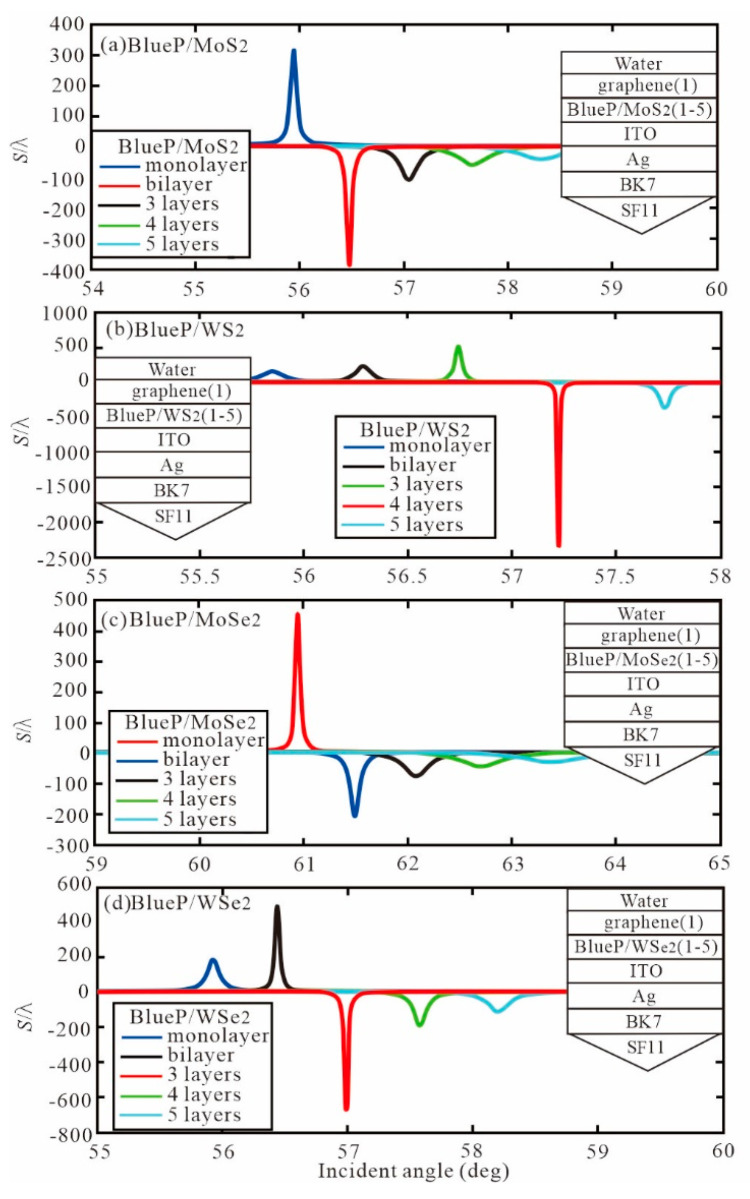
The GH shift with incident angle with graphene monolayer and different number of BlueP/TMDCs (**a**) BlueP/MoS_2_ layers, (**b**) BlueP/WS_2_ layers, (**c**) BlueP/MoSe_2_ layers, (**d**) BlueP/WSe_2_ layers.

**Figure 6 sensors-20-03605-f006:**
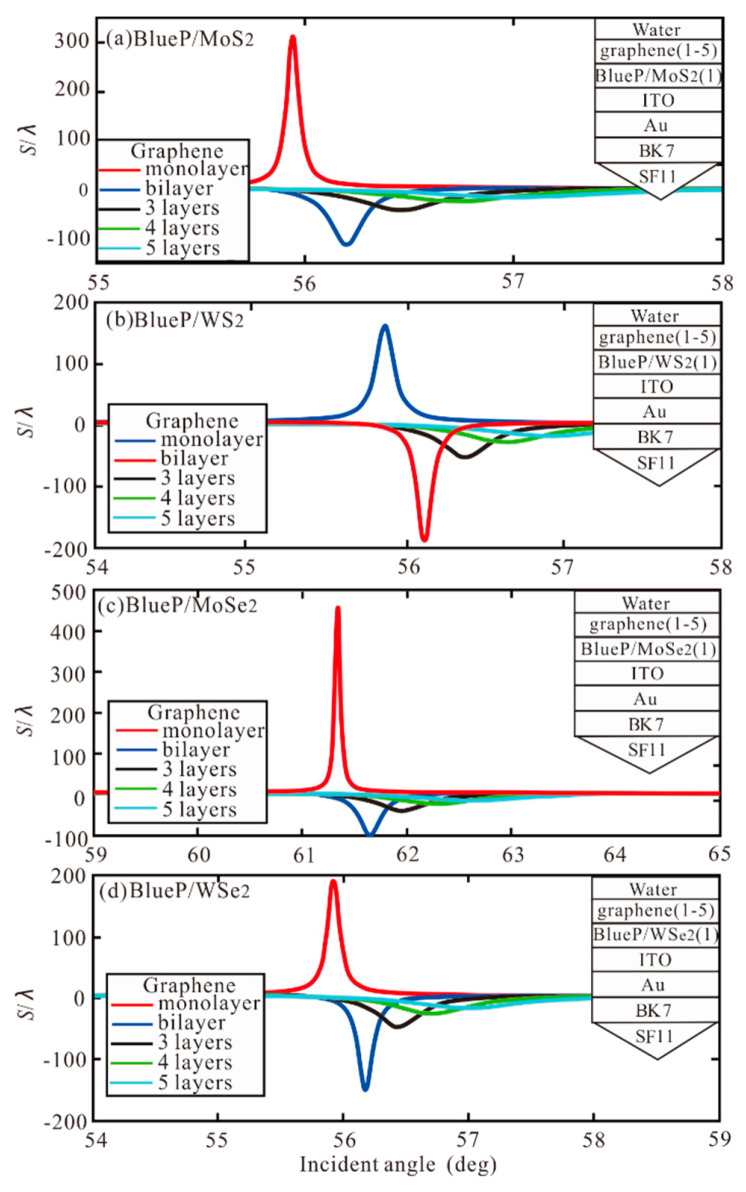
The GH shift with incident angle with BlueP/TMDCs monolayer and different number of graphene layers (**a**) BlueP/MoS_2,_ (**b**) BlueP/WS_2_, (**c**) BlueP/MoSe_2_, (**d**) BlueP/WSe_2_.

**Figure 7 sensors-20-03605-f007:**
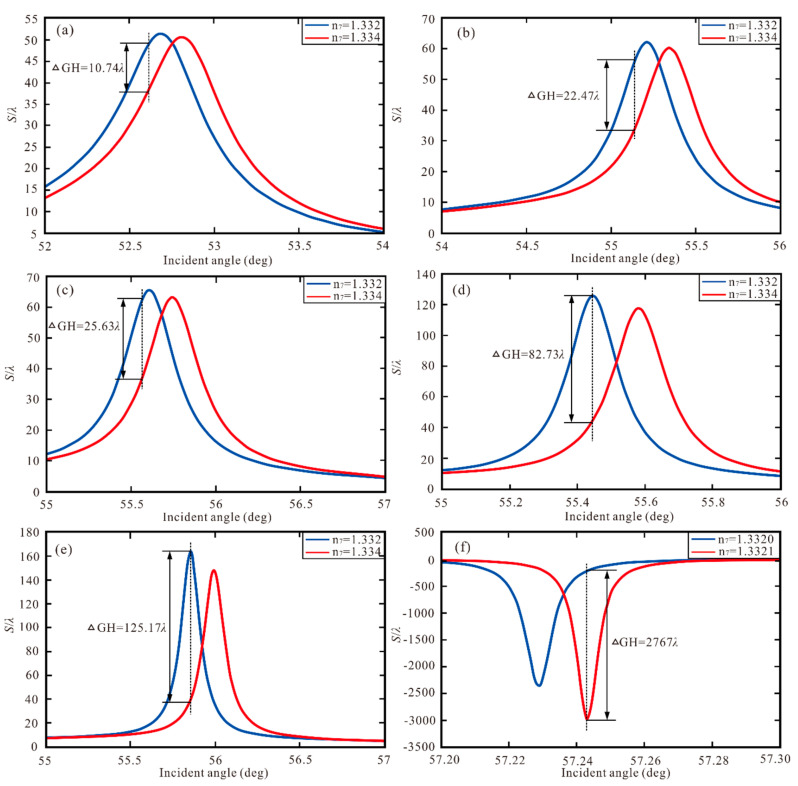
(**a**–**e**) GH shift with Structure I/Structure II/Structure III/Structure IV/Structure V with the Δn_7_ = 0.002, (**f**) GH shift with Structure VI with the Δn_7_ = 0.0001.

**Table 1 sensors-20-03605-t001:** The monolayer and refractive index (RI) of Blue Phosphorene (BlueP)/transition metal dichalogenides (TMDCs) and graphene at *λ* = 632.8 nm.

BlueP/TMDCs and Graphene	Monolayer (nm)	RI
BlueP/WS_2_	0.75	2.48 + 0.170*i*
BlueP/MoS_2_	0.75	2.81 + 0.320*i*
BlueP/MoSe_2_	0.78	2.77 + 0.350*i*
BlueP/WSe_2_	0.78	2.68 + 0.220*i*
graphene	0.34	3.00 + 1.149*i*

**Table 2 sensors-20-03605-t002:** The optimal GH shift (S/λ) with different number of BlueP/TMDCs.

BlueP/TMDCs	Monolayer	Bilayer	3 Layers	4 Layers	5 Layers
BlueP/MoS_2_	77.31	111.6	241.2	−662.2	−139.5
BlueP/WS_2_	65.69	70.32	76.23	83.68	92.83
BlueP/MoSe_2_	82.42	141.5	1188	−151.6	−65.77
BlueP/WSe_2_	68.57	78.32	93.58	119.1	166.3

**Table 3 sensors-20-03605-t003:** The optimal GH shift (S/λ) and resonance angle (*θ*) with different number of BlueP/TMDCs and graphene monolayer.

BlueP/TMDCs		Graphene (Monolayer)	
		GH Shift (λ)	Resonance Angle (*θ*)
BlueP/MoS_2_	bilayer	−385.8	56.48°
BlueP/WS_2_	4 layers	−2361	57.23°
BlueP/MoSe_2_	monolayer	456.9	55.96°
BlueP/WSe_2_	3 layers	−665.5	56.99°

**Table 4 sensors-20-03605-t004:** The optimal GH shift (S/λ) with BlueP/TMDCs monolayer and different number of graphene.

Graphene		BlueP/TMDCs (Monolayer)			
		BlueP/MoS_2_	BlueP/WS_2_	BlueP/MoSe_2_	BlueP/WSe_2_
**Graphene**	monolayer	315.6	164.1	456.9	193.1
	bilayer	−112.2	−189.3	−205.1	−153.6
	3 layers	−42.44	−52.23	−75.18	−48.25
	4 layers	−24.35	−27.95	−43.27	−26.49
	5 layers	−16.19	−18.02	−29.2	−17.27

**Table 5 sensors-20-03605-t005:** The optimal GH shift (S/λ) with different number of BlueP/TMDCs and graphene layers.

Type of BlueP/TMDCs and Graphene		Graphene				
		Monolayer	Bilayer	3 Layers	4 LAYERS	5 layers
**BlueP/MoS_2_**	bilayer	−385.8	−59.85	−29.7	−18.58	−12.9
BlueP/WS_2_	4 layers	−2361	−64.34	−29.65	−18.09	−12.4
BlueP/WSe_2_	3 layers	−665.5	−60.99	−29.16	−18.01	−12.43
BlueP/MoSe_2_	monolayer	456.9	−205.1	−75.18	−43.27	−29.2

**Table 6 sensors-20-03605-t006:** The optimal *S’_P_* with Δ*GH* and ∆*n*_7_ for Structure I to Structure VI.

Structure	ΔGH (λ)	∆*n*_7_	*S’_P_*(λ/RIU)
Structure I	10.74	0.002	5370
Structure II	22.4	0.002	11,200
Structure III	25.63	0.002	12,815
Structure IV	82.73	0.002	41,365
Structure V	125.17	0.002	62,585
Structure VI	2767	0.0001	2.767 × 10^7^

**Table 7 sensors-20-03605-t007:** Comparison with the previous studies GH shift and sensitivity SPR sensor.

Structure	GH shift(λ)	Sensitivity(λ/RIU)	References
Au	12.5	-	[9]
Air-MoS_2_	40.5	-	[50]
Air–graphene	61.1	-	[51]
Au-MoS_2_–graphene	235.8	5.545 × 10^5^	[36]
Au-ITO-MoSe_2_–graphene	801.7	8.02 × 10^5^	[37]
Ag-ITO-BlueP/WS_2_–graphene	2361	2.767 × 10^7^	This work

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
