# Peer review of "High-Sensitivity Goos-Hänchen Shifts Sensor Based on BlueP-TMDCs-Graphene Heterostructure"

_sensors, 2020, doi:10.3390/s20123605_

Round 1

Reviewer 1 Report

In this paper, the hybrid structure of BlueP/TMDCs (Blue/WS2, BlueP/MoS2, BlueP/MoSe2,73 BlueP/WSe2) and graphene coated with ITO and Ag thin film is propsed.
The description of Design consideration and mathematical model
 results, discussion should be more if this paper was submitted for a regular issue without page limit. Specially figures  need appropriate discussion and deteriorating the interesting results even though I recommend strongly the revision of manuscript. A notation list is necessary. While the simple conclusion is not adequate from an immediate technological perspective, a more detailed discussion/conclusion is in order particularly in the context of any other relevant reported studies of similar or dissimilar materials. Sufficient information is not included or cited to support assertions made and conclusions drawn. Discussion and conclusion need to be enhanced satisfactory.

Reviewer 2 Report

This manuscript theoretically investigates the Goos-Hänchen (GH) shifts in Ag-ITO-BlueP/TMDCs-graphene heterostructure for application in SPR biosensor. Moreover, to gain the highest GH shift, Kretschmann structure containing silver-indium tin oxide, graphene, and Blue phosphorene/transition metal oxides (BlueP/MoS2, BlueP/MoSe2, BlueP/WS2, and BlueP/WSe2) were calculated by the transfer matrix method and Fresnel equation. This topic is interesting but call for further improvements. My concerns about this work are as below:

  1. Usually there is a phase jump of up to 180 degree around a resonance. Why is the phase change in Figure. 2(a) over 180 degree?
  2. In the introduction part, “SPPs is a kind of vertically constrained evanescent electromagnetic wave” , the author should mention the latest literature, such as: Renewable Energy 158 (2020) 227-235; Solar Energy, 204 (2020) 635-643.
  3. Why is ITO added to the hybrid structure of SPR sensor? What role does ITO play in the sensor?
  4. Please explain the reason why SF11+BK7 glass are used in multi-layer structure in text.
